# TriMap: Large-scale Dimensionality Reduction Using Triplets

## Abstract

We introduce "TriMap"; a dimensionality reduction technique based on triplet constraints that preserves the global accuracy of the data better than the other commonly used methods such as t-SNE, LargeVis, and UMAP. To quantify the global accuracy, we introduce a score which roughly reflects the relative placement of the clusters rather than the individual points. We empirically show the excellent performance of TriMap on a large variety of datasets in terms of the quality of the embedding as well as the runtime. On our performance benchmarks, TriMap easily scales to millions of points without depleting the memory and clearly outperforms t-SNE, LargeVis, and UMAP in terms of runtime.

## 1 Introduction

Data visualization based on dimensionality reduction (DR) is a core problem in data analysis and machine learning. The aim of DR is to provide a low-dimensional representation (typically in 2D or 3D) of a given high-dimensional dataset that preserves the overall structure of the data as much as possible. The earlier approaches for DR involve linear methods such as PCA (Pearson, 1901). PCA aims to maintain the second-order statistics of the data by projecting the points into the low dimensional space that preserves the maximum amount of variance among all such projections. As a result, PCA has been shown to be effective in preserving the *global structure* of the data (Silva & Tenenbaum, 2003). The global structure includes the overall shape of the dataset, placement of the clusters, and existence of potential outliers. Unlike PCA, much of the focus of the more recent non-linear methods including t-SNE (Maaten & Hinton, 2008), LargeVis (Tang et al., 2016), and UMAP (McInnes et al., 2018) has been on preserving the local neighborhood structure of each individual point. Similarly, the common performance measures of DR such as trustworthiness-continuity (Venna & Kaski, 2005), precision-recall (i.e. AUC) (Venna et al., 2010), and nearest-neighbor accuracy have also been developed by retaining the same focus on reflecting the local accuracy of the embedding. Thus, there has been a lack of attention on developing methods that focus on preserving the global structure of the data and likewise, practical performance measures to assess the global accuracy.

We first introduce the *global score*, a quantitative measure which reflects the closeness of a given embedding to the PCA embedding (which is optimal by means of preserving the data variance). The purpose of this score is to measure the accuracy of an embedding in reflecting the overall placement of the clusters of points relative to their original representation in high-dimension. By design, PCA yields the highest global score among all the DR methods and high values of global score indicates the efficacy of a DR method in reflecting the global structure.

Next, we introduce *TriMap*, a DR method which focuses on preserving the global structure of the data in the embedding. Pairwise (dis)similarities between points (used by the previous DR methods) seem to be insufficient in capturing the global structure. Instead, TriMap incorporates a higher order of structure to construct the embedding by means of *triplets*:

$$(i, j, k) \Leftrightarrow \text{point } i \text{ is closer to point } j \text{ than point } k.$$

The key idea behind TriMap stems from semi-supervised metric learning (Amid et al., 2016): Given an initial low-dimensional representation for the data points, the triplet information from the high-dimensional representation of the points is used to enhance the quality of the embedding. Similarly, TriMap is initialized with the low dimensional PCA embedding, and this embedding is then modified using a set of carefully selected triplets from the high-dimensional representation.

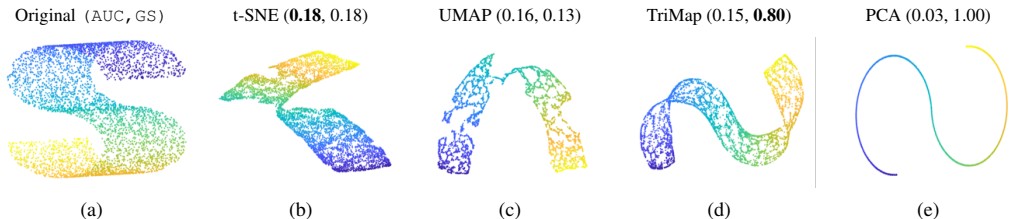

Figure 1: 2-D Visualizations of the S-curve dataset: (a) original dataset in 3-D, (b) t-SNE, (c) UMAP, (d) TriMap, and (e) PCA. The values of AUC and global score, for respectively measuring local and global accuracy, are shown in order as a pair (AUC, GS) for each embedding. Despite having higher AUC values, t-SNE and UMAP both fail to reflect the overall shape of the S-curve. On the other hand, TriMap successfully unveils the underlying structure in the original dataset. Note that GS is the only DR performance measure that can reflect this property.

With an extensive set of experiments, we show that TriMap produces excellent results on a variety of real-world as well as synthetic datasets. We show that in many cases TriMap outperforms all the competitor non-linear methods by means of global score and provides comparable local accuracy. While being significantly faster than t-SNE, TriMap provides comparable runtime to UMAP and LargeVis while scaling drastically better to larger datasets. On the Character Font Images dataset of ~1.7M points, TriMap calculates the embedding in ~1.3 hours while LargeVis takes more than 3 hours and UMAP exceeds the 12 hours time limit. Our contributions can be summarized as follows:

- We introduce a global score to quantify the quality of a low-dimensional embedding in reflecting the global structure of the high-dimensional data such as placement of the clusters rather than the local neighborhood of individual points.

- We introduce TriMap, a fast dimensionality reduction method which provides embeddings of the data that are globally more accurate than other non-linear DR methods such as t-SNE, LargeVis, and UMAP.

- We provide an efficient implementation[1] of TriMap that can easily scale to millions of points on commodity hardware and outperforms the competing methods in terms of runtime. We also perform many large-scale experiments on various datasets to show the efficacy of TriMap in terms of DR performance measures and runtime.

## 2 A MEASURE OF GLOBAL ACCURACY

Consider the S-curve dataset[2] which consists of 5000 points in 3-D uniformly sampled from an S-shaped manifold (Figure 1(a)). This dataset serves as a paradigmatic problem for evaluating the performance of DR methods. In Figure 1, we show the results of 2-D embeddings of the S-curve dataset using t-SNE, UMAP, TriMap, and PCA. The top of each graph is labeled by the scoring pair (AUC, GS) where GS stands for global score (introduced below). Note that both t-SNE and UMAP provide higher values of the AUC score and locally preserve the continuity of the manifold. However, they both fail to recover the global structure of the S-curve, which is naturally reflected in the PCA embedding. On the other hand, our TriMap method (formally defined later) successfully recovers the structure of the S-curve by "unveiling" the curved shape of the manifold at both ends. Overall, the 2-D TriMap embedding resembles the original 3-D representation as much as possible. Note also that GS is the only measure that can reflect the global accuracy of the embedding.

The previous example indicates that the local measures of DR performance (such as AUC) cannot reflect the global accuracy of a low-dimensional embedding. In fact, the low-granular structure of the data can only be estimated by considering the global statistics of the dataset, as regarded by the PCA method. PCA is a linear DR method that projects the high-dimensional data onto the top-$d$ orthogonal directions having the highest variance. In order to calculate the mapping, PCA only considers the aggregate statistics of the dataset rather than the local information of each individual data point. As a result, PCA is extremely well suited at retaining the *global structure* of the data, i.e. the overall shape of the dataset, placement of the clusters, and existence of potential outliers.

---

[1]https://github.com/ANONYMOUS
[2]https://scikit-learn.org/stable/modules/generated/sklearn.datasets.make_s_curve.html

NIL (NN = 0.906, GS = 0.927)    $\gamma = 50$ (NN = 0.935, GS = 0.924)    $\gamma = 500$ (NN = 0.940, GS = 0.916)    $\gamma = 5000$ (NN = 0.941, GS = 0.908)

• 0 • 1 • 2 • 3 • 4 • 5 • 6 • 7 • 8 • 9

      (a)                 (b)                 (c)                 (d)

Figure 2: The Effect of the weight transformation on the MNIST dataset: (a) no weight transformation, (b) $\gamma = 50$, (c) $\gamma = 500$ (default), and (d) $\gamma = 5000$. The values of nearest neighbor accuracy and global score are shown as a tuple (NN, GS) on top of each figure. Larger values of $\gamma$ emphasizes more on the local accuracy rather than the global accuracy.

However, by focusing on the global structure, PCA loses much of the local information such as the neighborhood structure of each data point.

Given a low-dimensional mapping produced by PCA, it is possible to calculate an optimal inverse mapping into the original high-dimensional space by means of minimizing the squared error. The optimal inverse map also corresponds to a linear mapping[3]. In order to quantify the global accuracy of a DR result, we focus on the accuracy of the embedding in reflecting the global structure of the data similar to PCA. That is, we consider the minimum reconstruction error of the original dataset by means of a linear inverse map. Given $n$ data points $\{\boldsymbol{x}_i \in \mathbb{R}^m\}_{i=1}^n$, let $\boldsymbol{X} \in \mathbb{R}^{m \times n}$ denote the high-dimensional data matrix where the $i$-th column corresponds to $\boldsymbol{x}_i$. Similarly, let $\boldsymbol{Y} \in \mathbb{R}^{d \times n}$ denote the matrix of the low-dimensional embedding of the points $\{\boldsymbol{y}_i \in \mathbb{R}^d\}_{i=1}^n$. Without loss of generality, we assume both $\boldsymbol{X}$ and $\boldsymbol{Y}$ are centered. We define the *Minimum Reconstruction Error (MRE)* from the embedding as

$$\mathcal{E}(\boldsymbol{Y}\,|\,\boldsymbol{X}) \coloneqq \min_{\boldsymbol{A} \in \mathbb{R}^{m \times d}} \|\boldsymbol{X} - \boldsymbol{A}\boldsymbol{Y}\|_{\mathrm{F}}^2\,,$$

where $\|\cdot\|_{\mathrm{F}}$ denotes the Frobenius norm[4]. Note that PCA has the lowest possible MRE among all the DR methods. Thus, in order to obtain a normalized measure of global accuracy of a given embedding $\boldsymbol{Y}$ for a data $\boldsymbol{X}$, we define the *global score (GS)* as

$$\mathrm{GS}(\boldsymbol{Y}\,|\,\boldsymbol{X}) \coloneqq \exp\left(-\frac{\mathcal{E}(\boldsymbol{Y}\,|\,\boldsymbol{X}) - \mathcal{E}_{\mathrm{PCA}}}{\mathcal{E}_{\mathrm{PCA}}}\right) \in [0, 1]\,,$$

where $\mathcal{E}_{\mathrm{PCA}} \coloneqq \mathcal{E}(\boldsymbol{Y}_{\mathrm{PCA}}\,|\,\boldsymbol{X})$ denotes the MRE achieved by the PCA embedding $\boldsymbol{Y}_{\mathrm{PCA}}$ on the same dataset $\boldsymbol{X}$. Note that $\mathrm{GS}(\boldsymbol{Y}_{\mathrm{PCA}}\,|\,\boldsymbol{X}) = 1$ and we claim that larger values of GS indicate a higher capacity of a DR method to reflect the global structure of the data, as shown in the experiments.

In the remainder of the paper, we use GS as the global measure of performance. Due to the high computational complexity for calculating the trustworthiness-continuity and AUC scores for large data sets, we use nearest-neighbors accuracy as the local measure of performance henceforth.

## 3  THE TRIMAP METHOD

We now formally introduce the TriMap method. Recall that a triplet consists of three points $(i, j, k)$ where point $i$ is closer to point $j$ than point $k$. TriMap chooses a subset $\mathcal{T} = \{(i, j, k)\}$ of triplets and assigns a weight $\omega_{ijk} \geq 0$ for each triplet: a higher value of $\omega_{ijk}$ implies that the pair $(i, k)$ is located much farther than the pair $(i, j)$. We define the loss of the triplet $(i, j, k)$ as

$$\ell_{ijk} \coloneqq \omega_{ijk} \frac{s(\boldsymbol{y}_i, \boldsymbol{y}_k)}{s(\boldsymbol{y}_i, \boldsymbol{y}_j) + s(\boldsymbol{y}_i, \boldsymbol{y}_k)}, \quad \text{where} \quad s(\boldsymbol{y}_i, \boldsymbol{y}_j) = \left(1 + |\boldsymbol{y}_i - \boldsymbol{y}_j\|^2\right)^{-1}\,,$$

---

[3]More specifically, mapping to low-dimension corresponds to projecting the data onto the top-$d$ eigendirections of the data covariance matrix. The inverse mapping is induced by the transpose of the projection matrix.

[4]The optimum value $\boldsymbol{A}^*$ for the MRE can be calculated efficiently as

$$\boldsymbol{A}^* = \boldsymbol{X}\boldsymbol{Y}^\top(\boldsymbol{Y}\boldsymbol{Y}^\top)^{-1}\,.$$

This also handles possible rotation and scaling of the embedding.

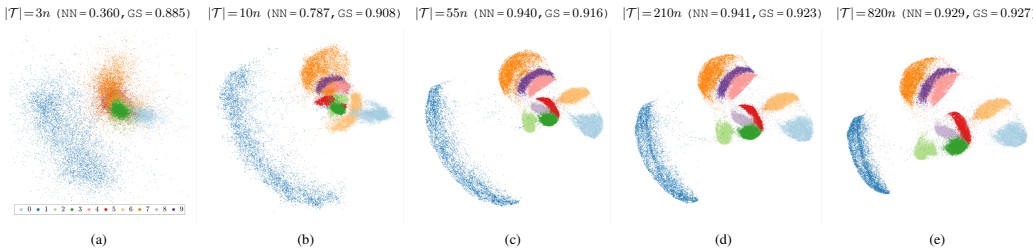

$|\mathcal{T}|=3n$ (NN=0.360, GS=0.885)  $|\mathcal{T}|=10n$ (NN=0.787, GS=0.908)  $|\mathcal{T}|=55n$ (NN=0.940, GS=0.916)  $|\mathcal{T}|=210n$ (NN=0.941, GS=0.923)  $|\mathcal{T}|=820n$ (NN=0.929, GS=0.927)

(a)           (b)           (c)           (d)           (e)

Figure 3: Effect of changing the number of triplets on the quality of the embeddings of the MNIST dataset. We consider $(m, m', r) = c \times (2, 1, 1)$ for: (a) $c = 1$, (b) $c = 2$, (c) $c = 5$ (default), (d) $c = 10$, and (e) $c = 20$. The values of nearest neighbor accuracy and global score are shown as a tuple (NN, GS) on top of each figure. The quality of embedding does not improve significantly after adding a certain number of triplets.

is a similarity function between $\boldsymbol{y}_i$ and $\boldsymbol{y}_j$. The choice of $s$ is motivated by the good performance of Student t-distribution for similarities in low-dimension in the t-SNE method. Note that the loss of the triplet $(i, j, k)$ approaches zero as $\|\boldsymbol{y}_i - \boldsymbol{y}_j\|$ decreases and $\|\boldsymbol{y}_i - \boldsymbol{y}_k\|$ increases.

We first develop the weighing scheme for the triplets. To reflect the relative similarities in high-dimension, we define the unnormalized weight of the triplet $(i, j, k)$ as

$$\tilde{\omega}_{ijk} = \exp(d_{ik}^2 - d_{ij}^2) \geq 0\,,$$

in which, $d_{ij}$ is any distance measure between $\boldsymbol{x}_i$ and $\boldsymbol{x}_j$ in high-dimension. For Euclidean distances, we use the scaling introduced in (Zelnik-Manor & Perona, 2005),

$$d_{ij}^2 = \frac{\|\boldsymbol{x}_i - \boldsymbol{x}_j\|^2}{\sigma_{ij}}\,,$$

where $\sigma_{ij} = \sigma_i \sigma_j$ and $\sigma_i$ is set to the average Euclidean distance between $\boldsymbol{x}_i$ and the set of nearest-neighbors of $\boldsymbol{x}_i$ from 4-th to 6-th neighbors. This choice of $\sigma_{ij}$ adaptively adjusts the scaling based on the density of the data.

Figure 4: $\gamma$-scaled log-transformation with different values of $\gamma$. The value NIL corresponds to no transformation.

While the choice of weights $\tilde{\omega}_{ijk}$ works well in practice, we adjust the weights further by applying a non-linear transformation that emphasizes the smaller weights. Expanding the values of small weights has the effect of placing the nearest-neighbors closer to the point and pushing the remaining points farther away, thus improving the local accuracy (as shown in Figure 2 and discussed later). The final value of the weight $\omega_{ijk}$ is obtained by applying the $\gamma$-scaled log-transformation (see Figure 4),

$$\omega_{ijk} = \zeta_\gamma\Big(\frac{\tilde{\omega}_{ijk}}{\mathcal{W}} + \delta\Big) \quad \text{where} \quad \zeta_\gamma(u) := \log\big(1 + \gamma\,u\big)\,,$$

in which $\mathcal{W} = \max_{(i',j',k')\in\mathcal{T}} \tilde{\omega}_{i'j'k'}$, $\gamma > 0$ is a scaling factor, and $\delta$ is a small constant. We use $\gamma = 500$ and $\delta = 10^{-4}$ in all our experiments.

To construct the embedding, we consider a small subset of all possible triplets $(i, j, k)$ for which, the closer point $j$ belongs to the set of nearest-neighbors of the point $i$ and the farther point $k$ is among the points that are more distant from $i$ than $j$, chosen uniformly at random. For each point we consider its $m = 10$ nearest neighbors and sample $m' = 5$ triplets per nearest-neighbor. This yields $m \times m' = 50$ nearest-neighbor triplets per point. In addition, we also add $r = 5$ random triplets $(i, j, k)$ per each point $i$ where $j$ and $k$ are sampled uniformly at random and their order is possibly switched based on their nearness to $i$. This yields $m \times m' + r = 55$ triplets per point in total. Thus, the overall complexity of the optimization step is linear in number of points $n$. The computational complexity is dominated by the nearest-neighbor search, which is shared among all the recent methods such as t-SNE, LargeVis, and UMAP. We use ANNOY for the approximate nearest-neighbor search[5] which is based on random projection trees.

While a random initialization for the embedding also works well in practice, we initialize the embedding to the PCA solution $\boldsymbol{Y}_{\text{PCA}}$ (scaled by a small constant value for better convergence). The PCA

---

[5]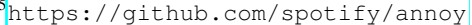 https://github.com/spotify/annoy

| Dataset (size) | t-SNE | LargeVis | UMAP | TriMap | Speedup |
|---|---|---|---|---|---|
| **COIL-20** (1440) | 00:00:08 | 00:05:51 | 00:00:04 | **00:00:02** | 2.00× |
| **USPS** (11K) | 00:02:02 | 00:06:12 | 00:00:12 | **00:00:11** | 1.10× |
| **Epileptic Seizure** (11.5K) | 00:03:11 | 00:06:17 | 00:00:15 | **00:00:12** | 1.25× |
| **20 Newsgroup** (18K) | 00:05:34 | 00:06:57 | 00:00:26 | **00:00:21** | 1.24× |
| **Tabula Muris** (54K) | 00:17:32 | 00:09:29 | 00:01:12 | **00:01:06** | 2.00× |
| **MNIST** (70K) | 00:20:38 | 00:11:29 | **00:01:15** | 00:01:23 | 0.90× |
| **Fashion MNIST** (70K) | 00:19:10 | 00:11:04 | **00:01:18** | 00:01:24 | 0.93× |
| **TV News** (∼129K) | 00:38:59 | 00:16:26 | 00:02:57 | **00:02:45** | 1.07× |
| **360+K Lyrics** (∼360K) | 08:50:49 | 00:44:16 | 00:25:23 | **00:13:49** | 1.84× |
| **Covertype** (∼581K) | – | 00:44:54 | 02:59:41 | **00:24:42** | 1.82× |
| **RCV1** (800K) | – | 01:34:38 | 04:55:53 | **00:36:59** | 2.56× |
| **Character Font Images** (∼1.7M) | – | 03:16:19 | – | **01:17:50** | 2.52× |
| **KDDCup99** (∼4.9M) | – | – | – | **04:17:01** | – |
| **HIGGS** (11M) | – | – | – | **10:08:36** | – |

Table 1: Runtime of the methods in `hh:mm:ss` format on single machine with 2.6 GHz Intel Core i5 CPU and 16 GB of memory. We limit the runtime of each method to 12 hours. Also, UMAP runs out of memory on datasets larger than ∼4M points.

initialization for TriMap allows faster convergence while preserving much of the global structure discovered by PCA. Note that the other DR methods such as t-SNE are extremely sensitive to the initialization and do not converge well with any initial solution other than small random initialization around the origin.

We define the final loss as the sum of the losses of the sampled triplets in $\mathcal{T}$

$$\ell_{\text{TriMap}} = \sum_{(i,j,k)\in\mathcal{T}} \ell_{ijk} \,.$$

The loss is minimized using the full-batch gradient descent with momentum using the delta-bar-delta method. In all our experiments, we perform 400 iterations with the value of momentum parameter equal to 0.5 during the first 250 iterations and 0.8 afterwards.

Finally, note that there exists connections between TriMap and a number of *triplet (aka ordinal) embedding* methods such as t-STE (Van Der Maaten & Weinberger, 2012). The triplet embedding methods have been developed for a different setting where the goal is to find an embedding based on a given pre-specified set of triplets obtained from human evaluators (or some form of implicit feedback). For instance, t-STE maximizes the sum of log of the satisfaction probabilities of the triplets to calculate the embedding. It is worth mentioning that TriMap is a DR method that is designed to sample the informative triplets from the high-dimensional representation of a set of points and assign weights to these triplets to reflect the relative similarities of these points. Although TriMap can also be used for the triplet embedding task, we only focus on the DR results [6].

## 3.1 EFFECT OF DIFFERENT PARAMETERS

We briefly discuss the effect of different parameters, namely the total number of triplets $|\mathcal{T}|$ and the $\gamma$-scaled log-transformation, on the quality of the embedding. TriMap is particularly robust to the number of sampled triplet for constructing the embedding. This can be explained by the high amount of redundancy in the triplets (the triplets $(i, j, k)$ and $(i, j, k')$ convey the same information if $k$ and $k'$ are nearest neighbors and also mapped nearby). In Figure 3 we consider various values for $m$, $m'$, and $r$ for the MNIST dataset while fixing the remaining parameters. In fact, using large number of triplets can sometimes introduce an overhead and require larger number of iterations to converge.

A more important parameter is $\gamma$ which controls the trade-off between the local and global accuracy. Larger values of $\gamma$ increases the relative importance of triplets with smaller weights. This causes the method to focus on the nearest-neighbor points rather than the points that are far away, thus improving the local accuracy. On the other hand, improving the local accuracy can impair the global accuracy. In Figure 2, we plot the $\gamma$-scaled log-transformation for various $\gamma$ values and illustrate the results on MNIST without the log-transformation as well as the results with different $\gamma$ values. For larger values of $\gamma$ the clusters tend to become more compressed and as a result, the nearest-neighbor accuracy is improved. On the other hand, the global score starts to decrease for larger $\gamma$ values.

---

[6]Also, the embeddings obtained by simply applying these methods to our set of sampled triplets are quite subpar (see the MNIST result in (Van Der Maaten & Weinberger, 2012)) and are not shown here.

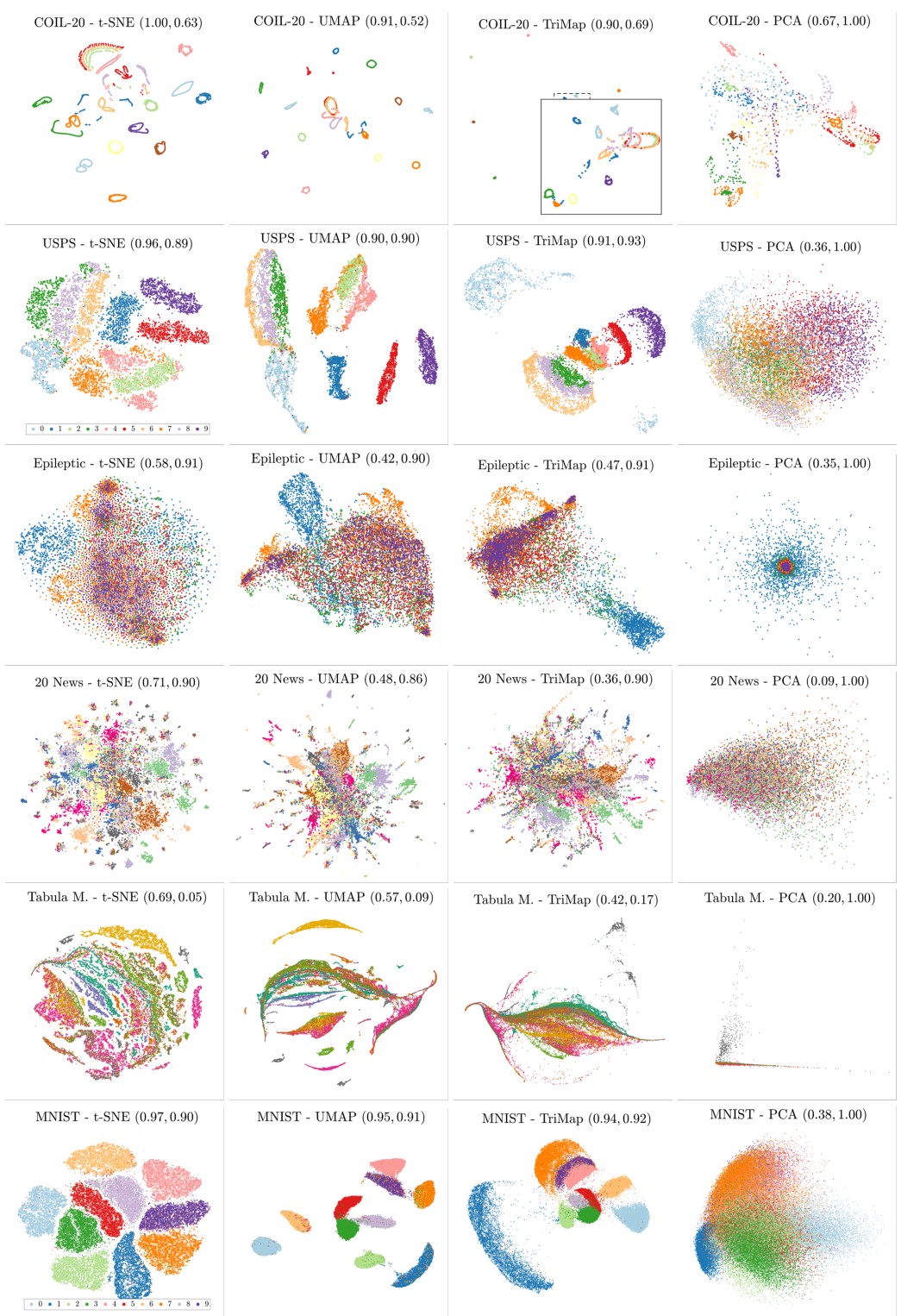

Figure 5: Visualizations of different datasets using t-SNE, UMAP, TriMap, and PCA. Each row corresponds to one dataset and each column represents one method. The values of nearest neighbor accuracy and global score are shown as a pair (NN, GS) on top of each figure.

# 4 EXPERIMENTS

In this section, we apply TriMap on a set of real-world as well as synthetic datasets and compare the results to t-SNE, LargeVis, UMAP, and PCA methods. The datasets used in our experiments are listed in Table 1 and a short description is given in the appendix. All experiments are conducted on

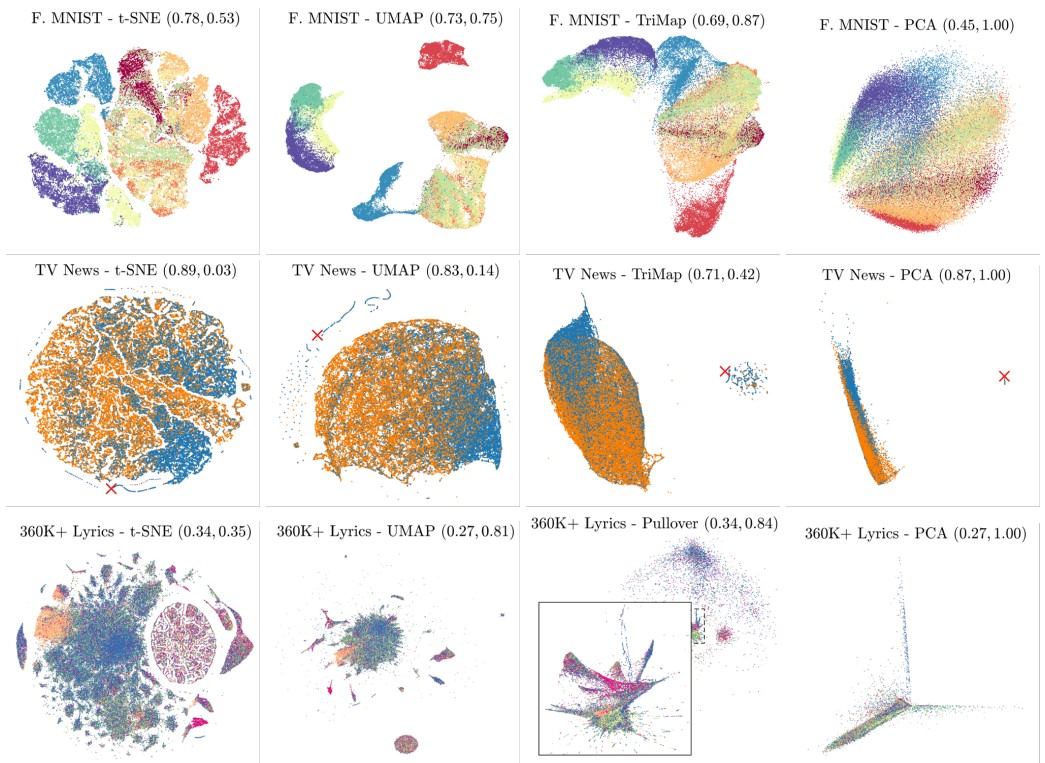

Figure 5: Visualizations of different datasets (continued) using t-SNE, UMAP, TriMap, and PCA. Each row corresponds to one dataset and each column represents one method. The values of nearest neighbor accuracy and global score are shown as a tuple (NN,GS) on top of each figure.

a single machine with 2.6 GHz Intel Core i5 CPU and 16 GB of memory. We limit the runtime of each algorithm to 12 hours. For implementations, we use the default `sklearn` implementation for t-SNE and the official implementations of LargeVis and UMAP provided by the authors[7],[8]. Due to lack of space, we provide the comparison to the LargeVis results as well additional TriMap results on the larger datasets in the appendix.

In order to have a fair comparison, we use the default parameter values for all methods, including ours ($m = 10$, $m' = 5$, $r = 5$, $\gamma = 500$, and 400 iterations). Also to reduce the overhead induced by the dimensionality of the data in the nearest-neighbor search step, we reduce the number of dimensions of the dataset to 100 if necessary, using the PCA method. To evaluate the local performance, we show the nearest-neighbor accuracy of each result. We also show the GS as a measure of global performance. The performance measures are shown on top of each figure as a pair (NN, GS).

## 4.1 RUNTIME

The runtime of the methods are provided in Table 1 in the `hh:mm:ss` format. We limit the runtime of each method to 12 hours. As can be seen from the results, TriMap provides excellent runtime and outperforms all the other methods in most cases. Also, TriMap easily scales to millions of points while the other methods exceed the time limit or run out of memory. For instance, UMAP causes an out of memory error for datasets larger than ∼4M points.

## 4.2 VISUALIZATIONS

The visualizations of the datasets using TriMap as well as the other competing methods are shown in Figure 5 and 6. For some results, we provide a zoomed in snippet over the main figure to provide a more detailed illustration. Overall, TriMap preserves the underlying global structure of the data better than the other competing methods. This is reflected by the larger GS values for TriMap as well as visually comparing the embeddings to the PCA result. For example, TriMap recovers the

---

[7]https://github.com/lferry007/LargeVis
[8]https://github.com/lmcinnes/umap

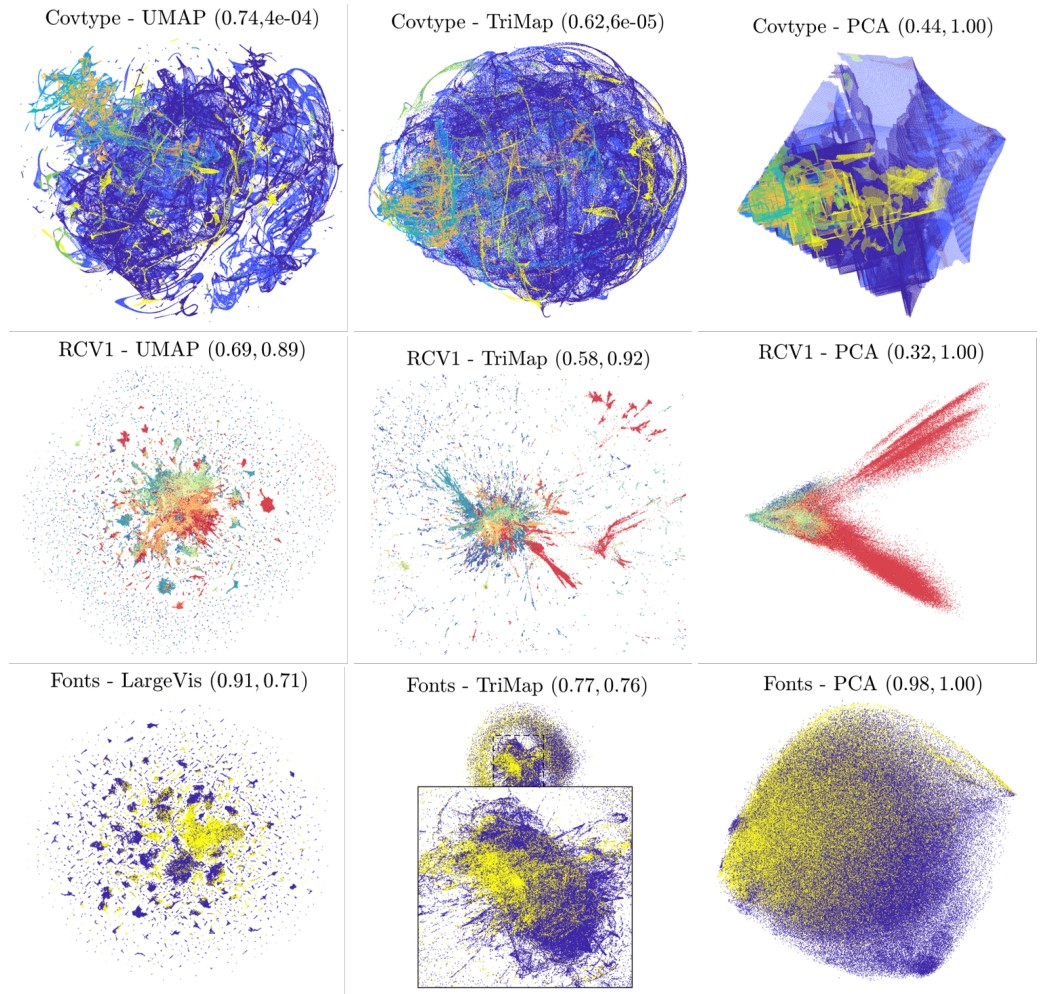

Figure 6: Visualizations of Covertype and RCV1 datasets using UMAP, TriMap, and PCA, and visualizations of the Character Font Images dataset using LargeVis, TriMap, and PCA. The values of nearest neighbor accuracy and global score are shown as a tuple `(NN,GS)` on top of each figure.

continuous structure of the TV news dataset and separates the remaining outliers in the data which are also identified by the PCA method. This can be verified by comparing the placement of an example outlier point, marked with a red ×, by the different methods: TriMap shows this point among other outliers whereas t-SNE and UMAP fail to uncover this information. Also, the global score of TriMap on this dataset is much higher than the other methods. Further discussion is given in the appendix.

## 5 CONCLUSION AND FUTURE WORK

TriMap is a fast and efficient method that can be easily applied to large datasets. While TriMap is extremely effective for uncovering the global structure of the data, other methods such as t-SNE can provide additional insight about the local neighborhood of individual points. As a future research direction, we consider using pairwise constraints along with triplet constraints to improve the local accuracy. The current implementation of TriMap utilizes a single core. Parallel implementation of the method that can exploit multiple cores is another future direction. Furthermore, the global accuracy is measured in terms of the global score which is based on the assumption that linear projection obtained by PCA is globally optimal. While our global score can provide insight about the global accuracy of the embedding in many cases, it appears to be ineffective when the data is highly non-linear or contains a large amount of outliers. Developing non-linear and more robust global performance measures could significantly improve the assessment of the DR results and provide guidelines for developing more accurate DR techniques.

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

## A DATASETS

The datasets used in the experiments are listed below. All datasets are publicly available online and a download link is provided.

- **COIL-20**[9] (1440): gray-scale images of 20 objects in uniformly sampled orientations (5 degrees of rotation, 72 images per object). Each image is pre-processed by having the background removed and cropped into size $128 \times 128$.
- **USPS**[10] (11K): images of handwritten digits (0–9) of size $16 \times 16$.
- **Epileptic Seizure**[11] (11.5K): EEG signal recordings of brain activity for seizure recognition. It contains 178-dimensional vectors belonging to 5 categories.
- **20 Newsgroup**[11] (18K): newsgroup posts categorized into 20 topics. We use a TF-IDF representation of the words in each document as the features.
- **Tabula Muris**[12] ($\sim$54K): single cell transcriptome data from the mouse from 20 organs.
- **MNIST**[13] (70K): images of handwritten digits (0–9) of size $28 \times 28$.
- **Fashion MNIST**[14] (70K): gray-scale images of clothing items such as t-shirt, pullover, bag, etc. of size $28 \times 28$.
- **TV News**[11] ($\sim$129K): audio-visual features from TV news broadcast categorized into commercial and non-commercial.
- **360K+ Lyrics**[15] ($\sim$362K): lyrics of songs from 12 different genres. We group similar genres together (metal-rock, R&B-pop, etc.) to form 7 groups. We use the TF-IDF representation of the words in the song as the features.
- **Covertype**[11] ($\sim$581K): cartographic features for forest cover type prediction.
- **RCV1**[16] (800K): Reuters Corpus Volume I archive of categorized newswire stories.
- **Character Font Images**[11] ($\sim$1.7M): images of character from scanned and computer generated fonts.
- **KDDCup99**[11] ($\sim$4.9M): computer network intrusion detection.
- **HIGGS**[11] (11M): Higgs bosons recognition from a background process.

## B MORE VISUALIZATIONS

We compare the results of TriMap to LargeVis in Figure 7 and 8. We also provide more visualizations obtained using TriMap in Figure 9.

## C DISCUSSION

We briefly discuss the results of TriMap and draw a comparison to the other methods.

TriMap generally provides better global accuracy compared to the competing methods. It also successfully maintains the continuity of the underlying manifold. This can be seen from the COIL-20 result where certain clusters are located farther away from the remaining clusters. However, the underlying structure for the main cluster resembles the one provided by the other methods. TriMap also preserves the continuous structure in the Fashion MNIST and the TV News datasets.

TriMap is also efficient in uncovering the possible outliers in the data. For instance, PCA reveals a large number of outliers in the Tabula Muris and the 360+K Lyrics datasets. These outliers are

---

[9] http://www.cs.columbia.edu/CAVE/software/softlib/coil-20.php
[10] https://www.kaggle.com/bistaumanga/usps-dataset
[11] http://archive.ics.uci.edu/ml/index.php
[12] https://tabula-muris.ds.czbiohub.org/
[13] http://yann.lecun.com/exdb/mnist/
[14] https://github.com/zalandoresearch/fashion-mnist
[15] https://www.kaggle.com/gyani95/380000-lyrics-from-metrolyrics
[16] https://scikit-learn.org/0.18/datasets/rcv1.html

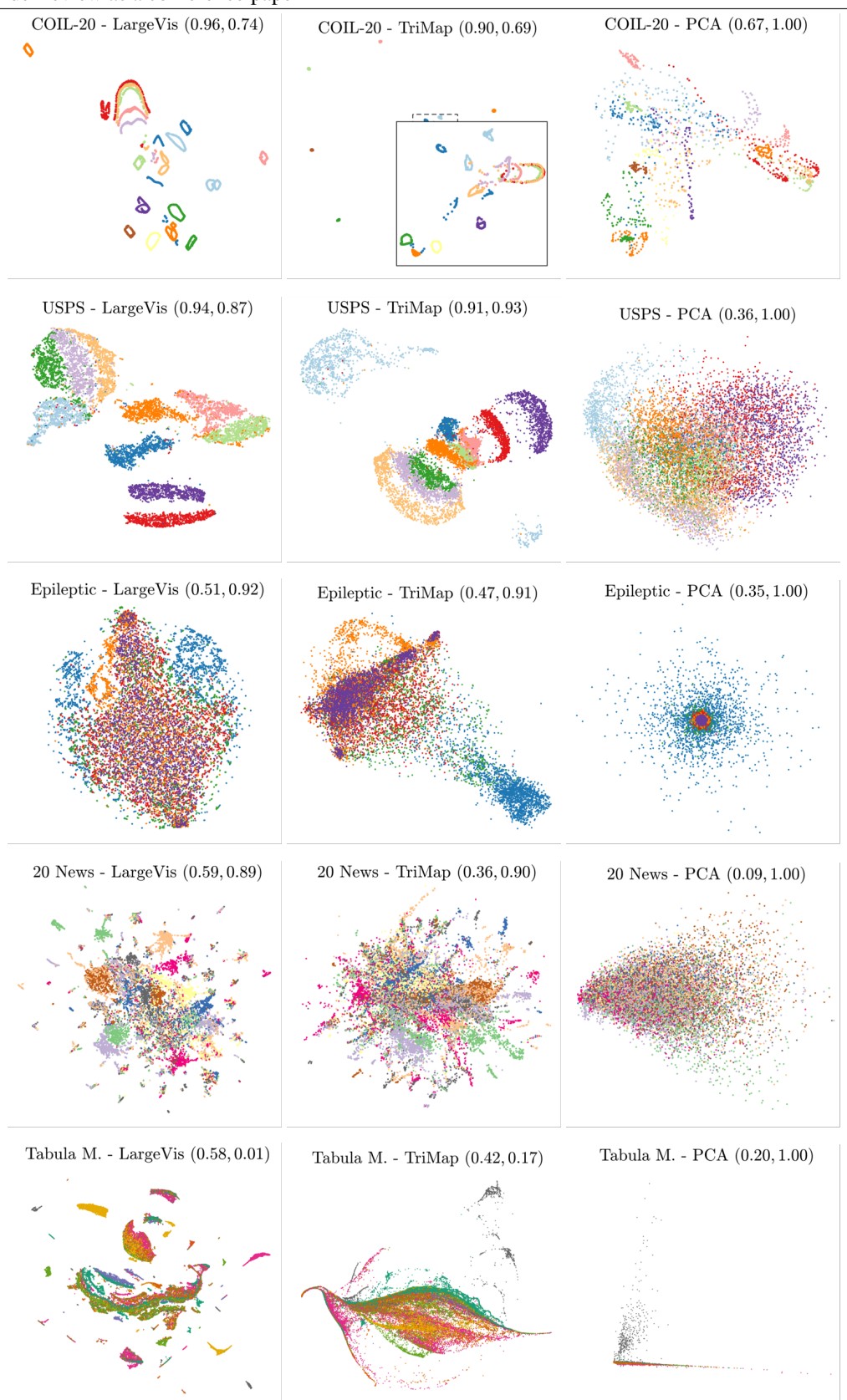

Figure 7: Visualizations of different datasets using LargeVis, TriMap, and PCA. Each row corresponds to one dataset and each column represents one method. The values of nearest neighbor accuracy and global score are shown as a pair `(NN,GS)` on top of each figure.

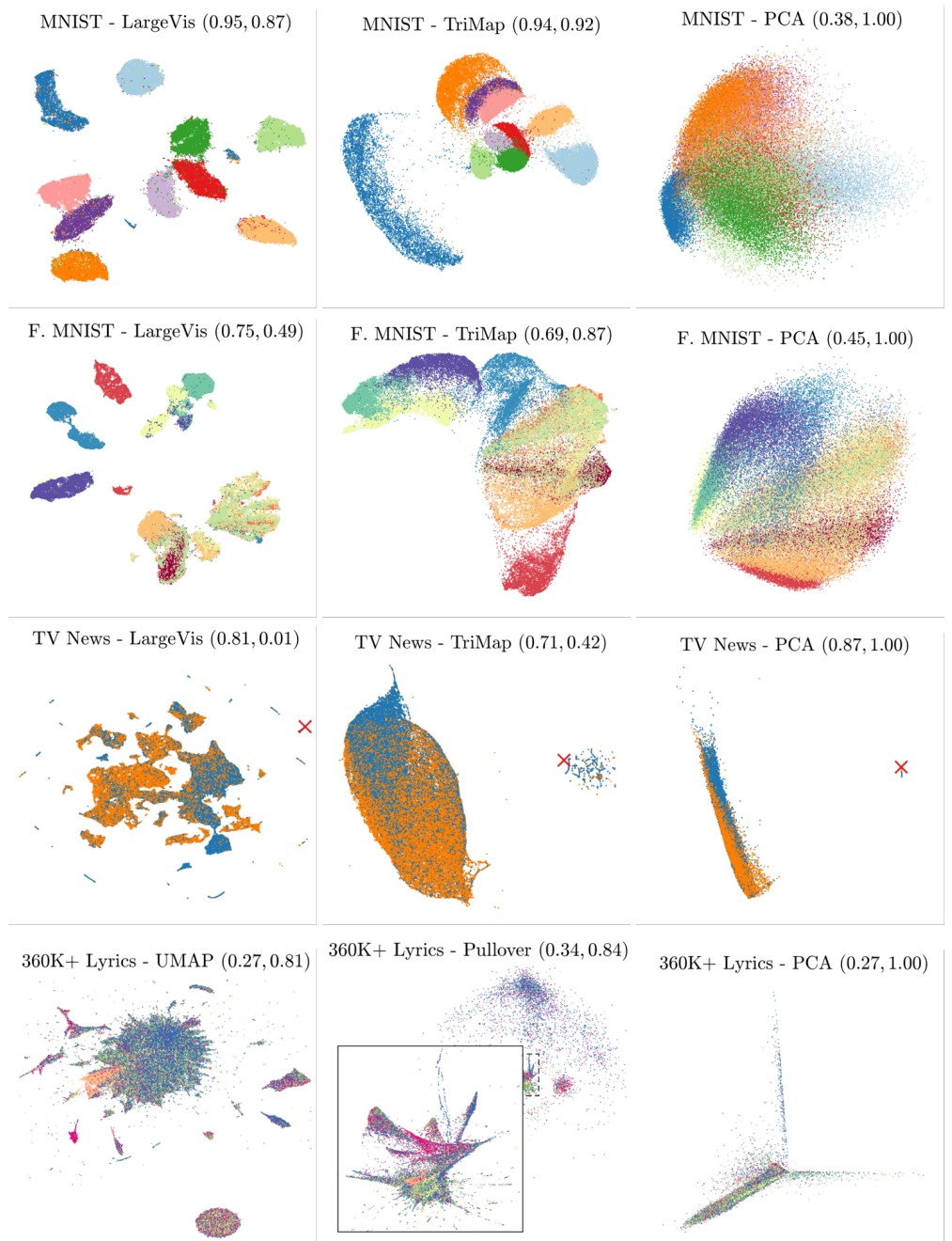

Figure 7: Visualizations of different datasets (continued) using LargeVis, TriMap, and PCA. Each row corresponds to one dataset and each column represents one method. The values of nearest neighbor accuracy and global score are shown as a tuple (NN,GS) on top of each figure.

located far away from the main clusters in the TriMap results. However, the same points are located very close to the remaining points in the t-SNE results.

Additionally, both t-SNE and LargeVis tend to form spurious clusters by splitting the underlying connected manifold. This can be seen from the TV News results and the result of LargeVis on the Covertype dataset.

Finally, notice that in some cases GS fails to reflect the global accuracy of the embeddings. This can be seen from the low GS values for all methods on the Covertype dataset. GS may become uninformative when there exists a high degree of non-linearity in the data that cannot be reflected using PCA. GS also cannot reflect the accuracy of the embedding in uncovering single outliers. Developing more accurate global measures for these scenarios is a future research direction.

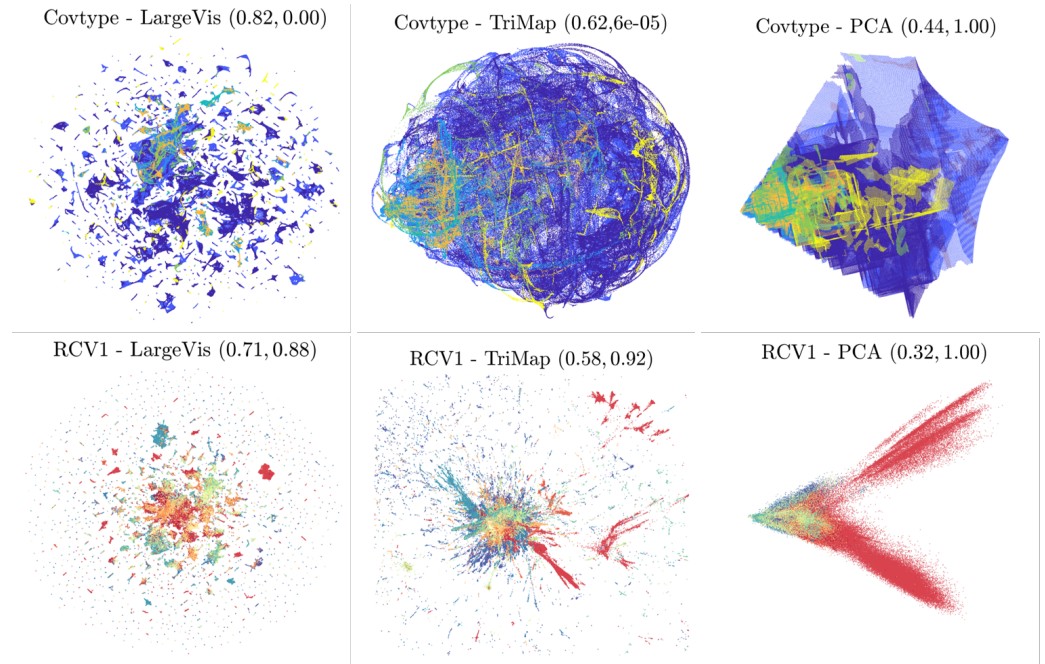

Figure 8: Visualizations of Covertype and RCV1 datasets using LargeVis, TriMap, and PCA, and visualizations of the Character Font Images dataset using LargeVis, TriMap, and PCA. The values of nearest neighbor accuracy and global score are shown as a tuple (NN,GS) on top of each figure.

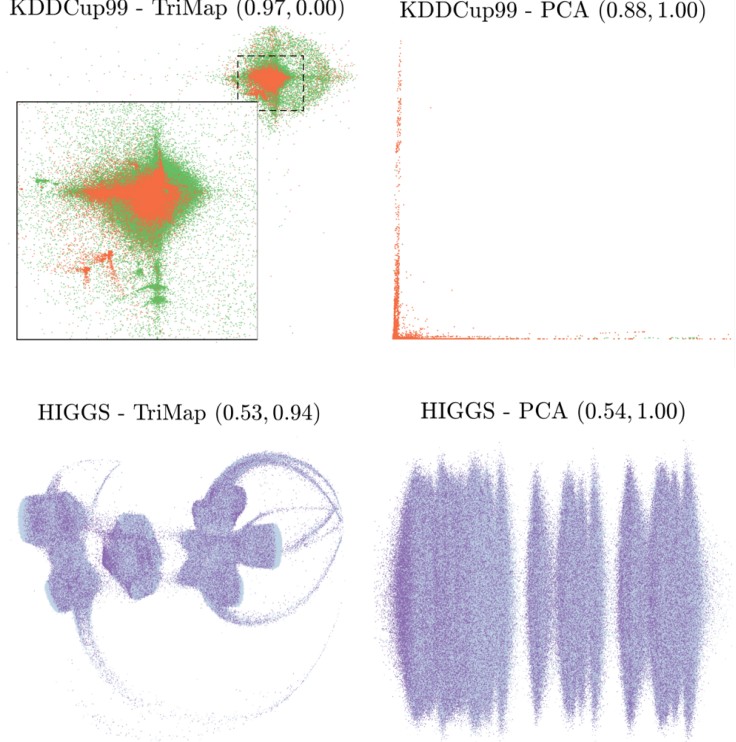

Figure 9: Visualizations of KKDCup99 and HIGGS datasets TriMap and PCA. Each row corresponds to one dataset and each column represents one method. The values of nearest neighbor accuracy and global score are shown as a tuple (NN,GS) on top of each figure. TriMap shows more structure for both datasets than PCA. Note that GS is uninformative for the KDDCup99 dataset.

