# OpenReview forum: "TriMap: Large-scale Dimensionality Reduction Using Triplets"
_ICLR.cc/2020/Conference — Reject_

### Official Review · AnonReviewer3 · 2019-10-15
**Official Blind Review #3**

**Rating:** 3

**Review:**

The paper proposed ``TriMap’’---a novel dimensionality reduction technique that learns to preserve relative distances among points in a triplet. The paper has done extensive experiments and presents the results in very nice visualizations. The paper is clearly written.

Major merits of this paper are:

1. The proposed method seems effective.

On some datasets (e.g., S-curve), the learned low-dimensional embeddings indeed look very good. And the proposed method has much less runtime than other baselines as shown in table-1.

2. The paper is well written.

However, I am still leaning towards rejecting this submission because the proposed method is lack of necessary justification.

1. Many important technical decisions on this method seem arbitrary, including the parametrization of functions (e.g., s, \omega, \zeta, etc) and values of hyperparameters (e.g., \gamma and \delta). For function forms, the authors should justify why particular parametrization has been chosen; for the hyperparameters, the authors should clearly explain how they are picked---maybe using domain knowledge or tuned on data?

2. The argument for ``global score’’ is not clear enough. There are at least two points that need clarification.

First, ``non-local’’ is not equal to ``global’’. The proposed method indeed considers non-local (or non-near) points while learning embeddings, which helps preserve non-local information. But I am not convinced that the preserved information is actually global. Maybe what helps is to first define ``global’’ in a dimensionality reduction context.

Second, the definition of ``global score’’ depends on another baseline method (i.e. PCA), which seems odd. A principled evaluation (or a score) should be method-independent. What seems right to me is to compute global score (i.e. how much global information has been preserved) by comparing to some statistics in the (high-dim) x space, not to another method.

Moreover, the authors had a strong claim that the proposed ``GS is the only DR performance measure that can reflect this property’’---it doesn’t sound right and why one can’t just use another score which is monotonic wrt the proposed score? The authors mentioned that ``PCA has the lowest possible MRE’’---but this is only right up to the use of a linear transformation and F-norm, so this shouldn’t be a justification for my questions above.

3. What is the reason for the successful runtime?

The authors didn’t clarify why the proposed method is theoretically faster than the baselines.

What I noted is: the authors chose a subset \mathcal{T} for the TriMap method and used \mathcal{T} throughout the paper---is it a typo or is a subset always chosen? If the latter holds, then how was it chosen and how large is it compared to the training data used by other methods? In the end, is the proposed method faster because it uses less data?

Besides the weakness above, I also suggest the authors evaluate their method with some extrinsic evaluations. What’s currently used is only intrinsic---the embeddings are trained to preserve relative distances and are evaluated on a trade-off between local accuracy and the defined global score. It is fine because extensive visualizations are provided and readers can subjectively judge the quality of the learned embeddings. However, the experimental section can be stronger if the authors can show the learned embeddings are better at helping some downstream tasks than other baselines (by preserving non-local information?).


**Experience Assessment:**

I do not know much about this area.

**Review Assessment: Checking Correctness Of Derivations And Theory:**

N/A

**Review Assessment: Checking Correctness Of Experiments:**

I carefully checked the experiments.

**Review Assessment: Thoroughness In Paper Reading:**

I read the paper at least twice and used my best judgement in assessing the paper.

---

> ### Author Response · Authors · 2019-11-12
> **The choice of parameters and the global score**
>
> Thanks for your comments.
>
> 1) We tried to justify each component in our method and show the reasoning for choosing each default value. In fact, most DR methods (t-SNE, LargeVis, UMAP) have multiple tunable parameters. Tuning the default values are mainly based on domain knowledge and the overall performance on a number standard datasets. Our default choice of parameters may not be optimal for every dataset. However, we show that TriMap has little sensitivity to choice of these parameters (please see Figures 2 and 3).
>
> 2) Although you make a valid point about global accuracy of DR methods, there has been no prior attempts to quantify these properties. Additionally, there exists no clear definition of the global accuracy. The global score, introduced in our paper, is in fact the first attempt to quantify the performance of a DR method in preserving the relative placements of the clusters. Note that it might be possible to define more complicated notions of global accuracy (for instance, using polynomial axis in the low-dimensional embedding to reflect the non-linearity). However, none of these approaches would be computationally feasible for very large datasets (~millions of points). Also, note that defining the global score in terms of PCA is equivalent to using second-order statistics (i.e. variance) of the data. Defining a global score based on other notions of distance than the Euclidean distance (i.e. Bregman PCA) is out of the scope of the current paper.
>
> Our claim that "GS is the only measure" means the only measure among the standard measures of DR (nearest-neighbor accuracy, trustworthiness-continuity, AUC, etc.). Indeed, any monotonically increasing transformation would have similar properties. We will clarify that sentence.
>
> 3) The subset of triplets \mathcal{T} is carefully sampled for each dataset. We show in Figure 3 that roughly a linear number of triplets for each dataset is sufficient to create the embedding. Please see the paragraph at the end of page 4 for a detailed description of the sampling procedure. The fast runtime of the method stems from the fact that it requires fewer computations than the other methods. The computational cost as well as the memory requirement of TriMap grows linearly after the initial NN search step. We are surprised that faster performance of the method is listed as the weakness of the method!

---

### Official Review · AnonReviewer2 · 2019-10-17
**Official Blind Review #2**

**Rating:** 1

**Review:**

Authors suggest a new technique for embedding point to low-dimensional space. The technique is reminiscent of t-SNE, with the difference that it get weighted triplets (i,j,k) as inputs (meaning that j is closer to i thank). Further a loss function is defined which directly follows t-SNE ideology.

A paper is purely experimental. The only way to judge the quality of its results is to compare 2D pictures of TriMap with other pictures. I did not see any evidence that images of TriMap somehow give a new insight into data (in comparison with other methods).

**Experience Assessment:**

I have published one or two papers in this area.

**Review Assessment: Checking Correctness Of Derivations And Theory:**

I assessed the sensibility of the derivations and theory.

**Review Assessment: Checking Correctness Of Experiments:**

I assessed the sensibility of the experiments.

**Review Assessment: Thoroughness In Paper Reading:**

I made a quick assessment of this paper.

---

> ### Author Response · Authors · 2019-11-12
> **Please provide a more thorough assessment of the paper**
>
> We argue that our method has a clear motivation. We justify the choice of each component and also provide empirical results along with standard quantitative measures to assess the performance of our method. Therefore, we believe the assessment of the paper should not be purely subjective. The two lines of comments provided by the reviewer do not address any of the contributions nor main points of discussion in the paper. We believe Reviewer #2 should at least expand their comments and provide a fair assessment of the paper.

---

### Official Review · AnonReviewer1 · 2019-10-23
**Official Blind Review #1**

**Rating:** 3

**Review:**

Authors introduce TriMap based on triplet constraints that preserves the global accuracy of the data. A measure of global accuracy is proposed to reflect the global accuracy of the embedding. Experiments on various datasets the better performance than baselines.

Authors define the minimum reconstruction error from the embedding as the global measure in reflecting the global structure of the data similar to PCA. From the definition, this measure has preferences to the linear projection model such as PCA, so the score becomes lower for non-linear projection methods such as t-SNE and UMAP.  The illustrated S-shape example in Figure 1 somehow demonstrate the difference of the proposed method with PCA, t-SNE and UMAP, but the usage of the embedding is not clear since Figure 1(d) looks like a 2-d visualizing of the original 3-d data visualized from certain angle. In addition, the initialization of the proposed method is the PCA method, which prefers the GS measure. It is interesting to see how the GS measure will change if the random initialization is used.

Authors demonstrate GS and AUS for all the tested data. It might be interesting and more important to see how to get the better embedding with a balanced score for a given data since GS and AUC seems always opposite measures.

The TriMap method defines the loss of triplet based on unnormalized weighting schema and the weights are adjusted by applying a non-linear transformation that emphasizes the small weights. These formulations are quite heuristic and constructive. It is better to have some formal explanation on the proposed method.


**Experience Assessment:**

I have published one or two papers in this area.

**Review Assessment: Checking Correctness Of Derivations And Theory:**

I assessed the sensibility of the derivations and theory.

**Review Assessment: Checking Correctness Of Experiments:**

I carefully checked the experiments.

**Review Assessment: Thoroughness In Paper Reading:**

I read the paper thoroughly.

---

> ### Author Response · Authors · 2019-11-12
> **TriMap is also a non-linear method like t-SNE and UMAP**
>
> Thank you for your comments.
>
> We would like to point out that TriMap is also a highly non-linear method. Please note that results of TriMap look significantly different than the linear projection results of PCA (for instance, see the Epileptic Seizure results). However, TriMap preserves "roughly" the same structure as the PCA result in most cases (e.g. MNIST and Fashion MNIST results). This includes relative placement of the clusters and existence of outliers (see the result on TV News). Please note that the global score by no means is the optimal score for assessing the performance of a DR method in terms of global accuracy. However, it can reflect the crude placement of the clusters to some extent.
>
> Empirically, we have tried random initialization and we did not observe significant difference in the final results. However, initialization to the PCA solution provides faster convergence (i.e. fewer number of iterations) to the final solution (as pointed out in the paper).
>
> Great point about balancing local and global measures! In fact we are considering this as a possible future direction. However, a careful tuning of the loss to strike a balance would be crucial. Nevertheless, the point of the current paper is to address some of the global properties that the other DR methods may fail to reflect.
>
> We tried to justify each component in our method and show the reasoning for choosing each default value. In fact, most DR methods (t-SNE, LargeVis, UMAP) have multiple tunable parameters. Tuning the default values are mainly based on domain knowledge and the overall performance on a number standard datasets. Most DR methods have some sort of heuristics for defining these components (for instance, the choice of the perplexity parameter in t-SNE). Other choices of these heuristics might also work in practice since DR is inherently an ill-posed and underdetermined problem.

---

### Decision · Program_Chairs · 2019-12-19

**Decision:**

Reject

**Comment:**

This paper proposes a new dimensionality reduction technique that tries to preserve the global structure of the data as measured by the relative distances between triplets. As Reviewer 1 noted, the construction of the TriMap algorithm is fairly heuristic, making it difficult to determine how TriMap ought to behave “better” than existing dimensionality reduction approaches other than through qualitative assessment. Here, I share Reviewer 2’s concern that the qualitative behavior of TriMap is difficult to distinguish from existing methods in many of the figures.